# Apolipoprotein-CIII *O*-Glycosylation, a Link between *GALNT2* and Plasma Lipids

**DOI:** 10.3390/ijms241914844

**Published:** 2023-10-02

**Authors:** Annemieke Naber, Daniel Demus, Roderick Slieker, Simone Nicolardi, Joline W. J. Beulens, Petra J. M. Elders, Aloysius G. Lieverse, Eric J. G. Sijbrands, Leen M. ’t Hart, Manfred Wuhrer, Mandy van Hoek

**Affiliations:** 1Department of Internal Medicine, Erasmus MC University Medical Center Rotterdam, P.O. Box 2040, 3000 CA Rotterdam, The Netherlands; a.naber@erasmusmc.nl (A.N.); e.sijbrands@erasmusmc.nl (E.J.G.S.); 2Center for Proteomics and Metabolomics, Leiden University Medical Center, P.O. Box 9600, 2300 RC Leiden, The Netherlands; daniel.demus@outlook.com (D.D.); s.nicolardi@lumc.nl (S.N.); m.wuhrer@lumc.nl (M.W.); 3Department of Cell and Chemical Biology, Leiden University Medical Center, P.O. Box 9600, 2300 RC Leiden, The Netherlands; r.c.slieker@lumc.nl (R.S.);; 4Department of Epidemiology and Data Science, Amsterdam UMC, Location Vrije Universiteit Amsterdam, P.O. Box 7057, 1007 MB Amsterdam, The Netherlands; j.beulens@amsterdamumc.nl; 5Amsterdam Public Health, Amsterdam Cardiovascular Sciences, Meibergdreef 9, 1105 AZ Amsterdam, The Netherlands; 6Julius Center for Health Sciences and Primary Care, University Medical Center Utrecht, P.O. Box 85500, 3508 GA Utrecht, The Netherlands; 7Department of General Practice, Amsterdam Public Health Institute, Amsterdam UMC, Location VUmc, P.O. Box 7057, 1007 MB Amsterdam, The Netherlands; p.elders@amsterdamumc.nl; 8Department of Internal Medicine, Maxima Medical Center, P.O. Box 90052, 5600 PD Eindhoven, The Netherlands; l.lieverse@mmc.nl; 9Department of Biomedical Data Science, Section Molecular Epidemiology, Leiden University Medical Center, Postal Zone S5-P, P.O. Box 9600, 2300 RC Leiden, The Netherlands

**Keywords:** apolipoprotein C-III, glycomics, genome-wide association study, hypertriglyceridemia, diabetes mellitus type 2

## Abstract

Apolipoprotein-CIII (apo-CIII) is involved in triglyceride-rich lipoprotein metabolism and linked to beta-cell damage, insulin resistance, and cardiovascular disease. Apo-CIII exists in four main proteoforms: non-glycosylated (apo-CIII_0a_), and glycosylated apo-CIII with zero, one, or two sialic acids (apo-CIII_0c_, apo-CIII_1_ and apo-CIII_2_). Our objective is to determine how apo-CIII glycosylation affects lipid traits and type 2 diabetes prevalence, and to investigate the genetic basis of these relations with a genome-wide association study (GWAS) on apo-CIII glycosylation. We conducted GWAS on the four apo-CIII proteoforms in the DiaGene study in people with and without type 2 diabetes (*n* = 2318). We investigated the relations of the identified genetic loci and apo-CIII glycosylation with lipids and type 2 diabetes. The associations of the genetic variants with lipids were replicated in the Diabetes Care System (*n* = 5409). Rs4846913-A, in the *GALNT2*-gene, was associated with decreased apo-CIII_0a_. This variant was associated with increased high-density lipoprotein cholesterol and decreased triglycerides, while high apo-CIII_0a_ was associated with raised high-density lipoprotein-cholesterol and triglycerides. Rs67086575-G, located in the *IFT172*-gene, was associated with decreased apo-CIII_2_ and with hypertriglyceridemia. In line, apo-CIII_2_ was associated with low triglycerides. On a genome-wide scale, we confirmed that the *GALNT2*-gene plays a major role i *O*-glycosylation of apolipoprotein-CIII, with subsequent associations with lipid parameters. We newly identified the *IFT172*/*NRBP1* region, in the literature previously associated with hypertriglyceridemia, as involved in apolipoprotein-CIII sialylation and hypertriglyceridemia. These results link genomics, glycosylation, and lipid metabolism, and represent a key step towards unravelling the importance of *O*-glycosylation in health and disease.

## 1. Introduction

Type 2 diabetes and cardiovascular disease are two of the leading causes of disease burden worldwide [1]. Triglycerides and low-density lipoprotein (LDL)-cholesterol (c) are important contributors to cardiovascular disease risk [2]. One of the central players in lipid metabolism is apolipoprotein-CIII (apo-CIII) [3].

Apo-CIII inhibits lipoprotein lipase [4] and hepatic lipase [5], which regulate triglyceride-rich lipoprotein (TRL) metabolism [6]. Apo-CIII is present mostly on TRLs when the triglyceride levels are high and mostly on high-density lipoproteins (HDL) when the triglyceride levels are low [7,8]. Apo-CIII levels and variation in the *APOC3* gene contribute to dyslipidaemia, artery wall inflammation, atherogenesis, and cardiovascular disease (CVD) risk [9,10,11]. Moreover, high apo-CIII levels have been linked to insulin resistance and beta-cell destruction [12,13]. *APOC3* promotor variants increase type 2 diabetes risk and the need for insulin treatment, particularly among lean patients [14].

Polypeptide N-acetylgalactosaminyltransferase 2 (GalNAc-T2), encoded by the GALNT2 gene, initiates mucin-type O-glycosylation of apo-CIII in the Golgi apparatus [15]. Mutations in the *GALNT2* gene cause rare congenital glycosylation disorders affecting apo-CIII glycosylation [16] and affect lipid profiles [17]. Without such mutations, apo-CIII exists in different proteoforms [18,19]. The four most abundant apo-CIII proteoforms are the non-glycosylated apo-CIII_0a_, and the proteoforms containing a mucin-type core-1 *O*-glycosylation with zero (apo-CIII_0c_), one (apo-CIII_1_), and two (apo-CIII_2_) sialic acids [18,19,20] (Appendix A). Apo-CIII proteoforms have different effects on hepatic TRL clearance receptors and the on potential of apo-CIII to inhibit lipoprotein lipase (LPL) resulting in variation in hepatic triglyceride clearance and lipid profile [21,22]. In smaller scale cohorts, the concentrations of apo-CIII_0a_, apo-CIII_0c_, and apo-CIII_1_ have been associated with lower small dense LDL [23] and higher triglyceride levels [18,24,25].

The objective of our study was to investigate the association between apo-CIII glycosylation and lipid traits, as well as the prevalence of type 2 diabetes, in a large population-based cohort. Moreover, we conducted the first genome-wide association studies (GWASs) of apo-CIII O-glycosylation in the DiaGene Study and explored the associations between genetic variants, apoCIII proteoforms, lipid levels, and type 2 diabetes. The genetic associations with lipid levels were subsequently replicated in the Hoorn Diabetes Care System cohort.

## 2. Results

### 2.1. Cohort Characteristics

Apo-CIII glycosylation was available for 2,318 participants in the DiaGene Study; the participants’ mean age was 65.3 (SD 9.5) years, 51% was female, the median body mass index (BMI) was 28.0 (IQR 25.2 to 31.6) kg/m^2^, 68% had type 2 diabetes, and 48% used lipid-lowering therapy. In the Hoorn DCS, genotypes of 5409 participants with type 2 diabetes were available; the mean age was 61.1 (SD 11.0) years, 44.6% was female, the median BMI was 29.4 (IQR 26.6 to 33.1) kg/m^2^ and 42.1% used lipid-lowering therapy (Table 1).

### 2.2. Apo-CIII O-Glycosylation, Type 2 Diabetes and Lipid Parameters

Apo-CIII_0a_ had a significant, negative association with the prevalence of type 2 diabetes (Table 2). In addition, apo-CIII_0a_ was significantly associated with high cholesterol levels from both HDL and non-HDL particles, including LDL. Apo-CIII_0c_ was associated with high triglycerides, total cholesterol, and non-HDL-c; but not with HDL-c or LDL-c. These associations point at more TRLs when the proportion of apo-CIII_0c_ is high. Apo-CIII_1_ and the apo-CIII_1_/apo-CIII_2_ ratio were positively and apo-CIII_2_ was negatively associated with LDL-c, non-HDL-c, total cholesterol, and triglycerides. HDL-c was associated with a lower apo-CIII_1_/apo-CIII_2_ ratio, lower proportion of apo-CIII_1_, and higher proportion of apo-CIII_2_ (Table 2). Adjustment for the use of lipid-lowering therapy and repeating these analyses for people with and without type 2 diabetes separately did not alter the direction of effect, although some associations lost significance (Appendix A).

### 2.3. GWASs of Apo-CIII O-Glycosylation Proteoforms

We conducted GWASs on the four main proteoforms of apo-CIII and the apo-CIII_1_/apo-CIII_2_ ratio in the DiaGene study. In total, we identified 11 SNPs with *p* < 1 × 10^−6^. The GWAS results are summarized in Table 3, Figure 1 and Appendix A. One signal, the A allele of rs4846913 at the *GALNT2* locus on chromosome 1, was genome-wide significantly associated with decreased apo-CIII_0a_ (*p* = 9.77 × 10^−36^) (Table 3, Appendix A). This SNP was also associated with high HDL-c (*p* = 0.065) and low triglycerides (*p* = 0.006) in the meta-analysis (Table 4), in line with public GWAS data (*p* = 7.46 × 10^−146^ [26] and 2.28 × 10^−235^ [27], respectively). The association of rs4846913 with apo-CIII_0a_ remained genome-wide significant when analysed for people with and without type 2 diabetes separately (Appendix A). The variant rs35498929-T, also in the *GALNT2* locus, was associated at a suggestive significance level with decreased apo-CIII_0a_ (*p* = 7.44 × 10^−7^) and with higher prevalence of type 2 diabetes (*p* = 0.006) in the relatively small DiaGene study for such analyses (Appendix A).

Two variants reached a suggestive significance level for associations with one or more apo-CIII proteoforms without reaching formal genome-wide significance but were also associated with one or more lipid traits in the meta-analyses. The rs10842926-C in the *PPFIBP1* locus, was associated with decreased apo-CIII_1_ (*p* = 7.07 × 10^−7^, Appendix A, Table 3). This variant was also associated with high LDL-c (*p* = 0.016) and low triglycerides (*p* = 0.039) with nominal significance in meta-analysis (Table 4) but was not significant in the Type 2 Diabetes Knowledge Portal [27]. The rs67086575-G allele, in the intronic region of the *IFT172*-gene was associated with apo-CIII_2_ and the apo-CIII_1_/apo-CIII_2_ ratio (*p* = 7.32 × 10^−07^ and *p* =7.37 × 10^−07^, respectively); and was significant for higher triglycerides (*p* = 1.07 × 10^−5^) in the meta-analysis, and for higher LDL-c, total cholesterol, and non-HDL-c (*p* = 0.019, *p* = 0.025, and *p* = 0.013, respectively) in the DiaGene study (Appendix A, Table 4). The associations with triglycerides (*p* = 4.84 × 10^−287^) [27], LDL-c (*p* = 2.90 × 10^−51^) [26], and total cholesterol (*p* = 4.76 × 10^−26^) [26] were in line with public data from the Type 2 Diabetes Knowledge Portal and the IEU OpenGWAS Project. The associations of our 11 identified genetic loci with apo-CIII proteoforms had the same direction of effect in people with and without type 2 diabetes (Appendix A). Other variants had a promising association with an apo-CIII proteoform and showed a significant association with clinical traits in one of the studies, but these signals did not survive the meta-analyses (in Appendix A).

## 3. Discussion

In the present study, we present the first GWAS of protein *O*-glycosylation, combined with clinical outcomes. We investigated the associations of apo-CIII proteoforms with lipid traits and prevalent type 2 diabetes, and performed GWASs on apo-CIII *O*-glycosylation to determine the direction of the associations. We found that the *GALNT2* locus was strongly associated with apo-CIII glycosylation and lipid traits. These findings were replicated in an independent population and then meta-analysed. The *GALNT2* locus linked apo-CIII_0a_ and triglyceride levels to prevalent type 2 diabetes. To our knowledge, this is the first paper describing a genome-wide association with apo-CIII glycosylation. Moreover, a variant at the *IFT172*/*NRBP1* region, not previously linked to apo-CIII glycosylation, showed consistent associations with apo-CIII sialylation, and with triglyceride and total cholesterol levels.

We confirmed on a genome-wide significant scale that the *GALNT2*-gene plays a major role in the *O*-glycosylation of apo-CIII. The *GALNT2*-gene encodes the GalNAc-T2 enzyme that links N-acetylgalactosamine (GalNAc) to proteins, which is the essential first step in the addition of mucine-type O-linked glycans [28]. Apo-CIII is a selective GalNAc-T2 target: it is dependent of GalNAc-T2 for its glycosylation [29]. In our study, the *GALNT2* variants rs4846913-A and rs35498929-T were associated with a low proportion of non-glycosylated apo-CIII_0a_, reflecting increased *O*-glycosylation of apo-CIII. The rs4846913 variant is located within a regulatory element that drives the expression of the *GALNT2*-gene in human hepatocytes [30], which is in line with our findings of increased apo-CIII glycosylation. In vitro studies suggest that the rs4846913-A allele increases binding with transcription factor cytosine-cytosine-adenosine-adenosine-thymidine (CCAAT) enhancer binding protein beta (CEBPB), possibly leading to increased expression of *GALNT2* [30], while the C-allele has low functional activity [31].

Rs4846913-A also had a negative association with triglyceride levels, in line with findings in European [26,27] and Asian consortia [32]. In contrast, Holleboom et al. found that rare *GALNT2* missense variants resulted in attenuated glycosylation of apo-CIII and better triglyceride clearance [22]. Holleboom et al. [22] found altered apo-CIII sialylation patterns in the carriers of a loss-of-function *GALNT2* variant. They speculated that this was due to a smaller amount of glycosylated apo-CIII available for sialylation, increasing the number of disialylated species. In our study, sialylation within *O*-glycosylated species (apo-CIII_0c_, apo-CIII_1_, and apo-CIII_2_) did not associate with the rs4846913 and rs35498929 *GALNT2* variants. Our genetic loci may not share the mechanism described by Holleboom et al. In our data, apo-CIII_0a_ had a small positive association with triglyceride levels, which is in line with findings by Yassine et al. [18]. Apo-CIII_0a_ was associated with higher HDL-c. Unfortunately, there is barely any literature about apo-CIII_0a_ and HDL-c. Koska et al. found opposite directions of effect in their two study cohorts for the association of apo-CIII_0a_ with HDL-c [24]. As demonstrated in vitro, the GalNAc-T2 enzyme, encoded by the *GALNT2*-gene, also glycosylates angiopoietin-like protein 3 (ANGPTL3), lecithin-cholesterol acyltransferase (LCAT), and phospholipid transfer protein (PLTP), which all play important roles in lipid metabolism, and especially triglyceride and HDL metabolism. Cholesteryl ester transfer protein (CETP) is another glycoprotein whose activity was directly associated with serum triglycerides and inversely with HDL-c [33]. We cannot exclude that *GALNT2* can affect lipid profiles through these or other proteins [29]. Moreover, to our knowledge, it is unknown whether glycosylation of these proteins parallels that of apo-CIII in type 2 diabetes. These aspects should be elucidated in future studies.

Regarding sialylation, apo-CIII_2_ and the apo-CIII_1_/apo-CIII_2_ ratio had strong negative and positive associations with triglyceride levels, respectively. Furthermore, rs67086575-G was associated with low apo-CIII_2_, high apo-CIII_1_/apo-CIII_2_ ratio, and with high triglycerides in the meta-analyses. The variant also had a positive association with LDL-c, non-HDL-c and total cholesterol but this was confined to the DiaGene study. The association of this variant with triglycerides, LDL-c and total cholesterol is in agreement with findings in public GWAS data [26,27]. Rs67086575 is located in the intronic region of the *IFT172*-gene. This gene encodes a peripheral subunit of the intraflagellar transport subcomplex-B (IFT-B) [34]. Mutations in this gene have been associated with ciliopathies, but so far not with glycosylation [34]. However, a link between primary cilia and obesity, insulin signalling, and type 2 diabetes have been described, which makes crosslinks to the lipoprotein metabolism not unlikely [35]. In blood cell lines, this SNP is associated with increased expression of the *NRBP1*-gene [36]. This *NRBP1*-gene is suggested to play a role in subcellular trafficking between the endoplasmic reticulum and Golgi apparatus [37]. Since sialylation of *O*-glycans takes place at the Golgi-membrane [38], the link between rs67086575 and the *NRBP1*-gene might explain the association of this SNP with apo-CIII sialylation.

Apo-CIII_2_ is associated with higher HDL-cholesterol, lower LDL-cholesterol, and lower triglyceride levels, while apo-CIII_0c_ and apo-CIII_1_ showed opposite effects. Our findings are in accordance with findings in type 2 diabetes and prediabetes by Koska et al. [24]. Moreover, glycosylation of apo-CIII may affect its inhibitory potential on LPL, although the direction and magnitude of this effect remains to be determined. Furthermore, apo-CIII glycosylation influences the interaction of LDL with the vascular wall [22,39]. From our data, we could speculate that lowering apo-CIII_2_ or increasing apo-CIII_1_ and apo-CIII_0a_ might cause a shift towards a more atherogenic lipid profile. These glycoforms might be an interesting treatment target for the prevention of cardiovascular disease in the general population and in type 2 diabetes. Future large-scale research is needed to confirm the effects of apo-CIII proteoforms on lipid metabolism and cardiovascular disease risk, and to investigate the effect of medication on apo-CIII proteoform distributions, which is relevant for the development and effectivity of lipid-lowering therapy.

Hepatic receptors differentially clear sialylated apo-CIII glycoforms [21], therefore it has to be considered that changed proportions of certain apo-CIII proteoforms might not only be the result of aberrant posttranslational modification of apo-CIII, but might also be the result of accumulation due to dysfunctional clearance pathways. Nevertheless, our genetic findings pointed mainly towards the posttranslational modification of apo-CIII as a cause of aberrant glycosylation and sialylation. Larger GWASs might elucidate possible changes in clearance pathways of apo-CIII proteoforms.

This is the first GWAS of apo-CIII *O*-glycosylation, and therefore provides new insights into the genetic background of *O*-glycosylation. Current knowledge regarding the genetics of apo-CIII *O*-glycosylation is mainly based on studies addressing congenital disorders of glycosylation, where rare genetic variants have been shown to affect apo-CIII *O*-glycosylation [16]. Findings in these disorders might not be applicable to the general population. Our findings had the same direction and magnitude of effect in people with and without diabetes, which suggests applicability to the general population. The DiaGene Study is the first cohort with detailed information on total plasma apo-CIII *O*-glycosylation on a large scale, using a high-throughput method to measure apo-CIII proteoforms. To date, the relations between lipids and apo-CIII glycoforms are only investigated in small cohorts. Our findings are in line with those studies, although non-glycosylated apo-CIII_0a,_ has received little attention in the current literature.

There are no previous large-scale studies with data on apo-CIII *O*-glycosylation. Therefore, direct replication of our GWAS is not possible; this limits our power to detect small effect sizes. Nevertheless, we could only detect associations with substantial effect sizes, which are most relevant as a starting point. Notably, we did replicate the identified lead SNPs from the GWAS with plasma lipids in the independent Hoorn DCS cohort to improve power for effects on the clinical endpoints and decrease the risk of false-positive results. The Hoorn DCS cohort included patients with type 2 diabetes treated by their general practitioner, whereas the DiaGene study included patients with type 2 diabetes from all lines of care and people without diabetes. This could explain some of the differences between these cohorts. A few associations had opposing directions of effect in the DiaGene and Hoorn DCS cohorts. However, the confidence intervals were mostly overlapping, which does not allow definite conclusion on the directions of the effect. Unfortunately, we do not have total apo-CIII plasma concentration data. The distribution of glycoforms in plasma is stable in variable apo-CIII concentrations in young, healthy men [40]. By adding the genetic basis to the analyses, we could offset this limitation for the most part. Both cohorts were mainly of Caucasian descent; therefore, we cannot generalise our findings to other ethnic groups.

In conclusion, variants of the *GALNT2*-gene affect apo-CIII *O*-glycosylation, lipid metabolism, and the risk of type 2 diabetes. The GALNT2-gene encodes the GalNAc-T2 enzyme that covalently links N-acetylgalactosamine (GalNAc) to proteins. This suggests that the order of causality has the *O*-glycosylation at its basis, subsequently affecting apo-CIII function and then altering the lipid profile and the risk of type 2 diabetes, and not vice versa. Moreover, we propose the *NRBP1*-gene as a possible player in the sialylation of apolipoprotein-CIII and hypertriglyceridemia. Our results confirm a link between genomics, glycosylation, and lipid metabolism. This is a key step towards unravelling the regulation of serum and plasma glycoprotein O-glycosylation in health and disease. Glycophenotype characterisation alongside genetic variant identification should serve as relevant prognostic and predictive tools and should be considered for target identification of new pharmacological agents.

## 4. Material and Methods

### 4.1. DiaGene Study

We used cross-sectional data from the DiaGene Study. This study has been described in more detail elsewhere [41]. Briefly, this case-control cohort comprises type 2 diabetes patients from all lines of care and people without diabetes from the area of Eindhoven, the Netherlands. We had available data on apo-CIII *O*-glycosylation for 1572 persons with type 2 diabetes and 746 persons without diabetes, of which, for 1872 participants, genetic data were also available to perform the GWAS analysis.

### 4.2. The Hoorn Diabetes Care System Cohort

For replication of the genetic associations with lipid levels, we used data from the Hoorn Diabetes Care System (Hoorn DCS) cohort, described in detail elsewhere [42]. Patients with type 2 diabetes treated in primary care in the region of West Friesland, the Netherlands, are included in this cohort. Genetic, biochemical, anthropometric, and clinical data of 5409 persons with type 2 diabetes were available. Apo-CIII glycosylation measurements were not available in the Hoorn DCS cohort.

Informed consent was obtained from all subjects involved in the study. Both studies were approved by the Medical Ethics Committees of the involved hospitals in compliance with the Declaration of Helsinki principles (DiaGene MEC-2004-230, Hoorn DCS 2007/57).

### 4.3. Apo-CIII Glycosylation Measurements

The analysis of apo-CIII *O*-glycosylation in the DiaGene Study was performed using a high-throughput method based on solid-phase extraction, matrix-assisted laser desorption/ionization (MALDI) and ultrahigh-resolution Fourier transform ion cyclotron resonance (FT-ICR) mass spectrometry (MS), described by Demus et al. [25]. The method provides the coefficient of variation values in a range of 6–16% for inter-plate and 1–18% for average intra-plate variability for relative quantitation of all four apo-CIII proteoforms. Apo-CIII_0a_ represents non-glycosylated apo-CIII; the non-, mono- and di-sialylated glycoforms are described as apo-CIII_0c_, apo-CIII_1_ and apo-CIII_2_, respectively. Apo-CIII_0a_ was normalized to the sum of all four proteoforms: apo-CIII_0a_, apo-CIII_0c_, apo-CIII_1_, and apo-CIII_2_ to reflect the proportion of non-glycosylated apo-CIII. The sum of apo-CIII_0c_, apo-CIII_1_, and apo-CIII_2_ was set to 1.0 to obtain the proportion of these three glycoforms within all glycosylated species of apo-CIII. Additionally, the ratio of apo-CIII_1_ and apo-CIII_2_ was calculated. Individuals with missing glycosylation data, missing age, sex, or genetic principal components were excluded from the analyses. Glycan measurements were natural log-transformed prior to GWAS analysis, because of the skewness of their distributions.

### 4.4. GWAS of Apo-CIII Glycosylation

In the DiaGene and Hoorn DCS cohorts, quality control was performed in PLINK [43], and genotypes were imputed to the Haplotype Reference Consortium r1.1 reference panel [44,45] using the Michigan Imputation Server [46]. Variants with an imputation quality < 0.4, minor allele frequency ≤ 0.05 or ≥0.95; effective allele count ≤ 5; or when out of Hardy–Weinberg Equilibrium (HWE) with HWE *p*-Value ≤ 1 × 10^−4^ were excluded. Using zCall [47] we aimed to call previously uncalled genotypes, mostly of rare variants. Manhattan plots, QQ plots, and locus zoom plots were generated using FUMA [48]. In the DiaGene study, we conducted separate GWASs on all four apo-CIII proteoforms, and the apo-CIII_1_/apo-CIII_2_ ratio with RVtest [49] and quality control was performed with EasyQC [50]. We assessed genetic associations using linear regression models adjusted for age, sex, and significant principal components of ancestry (Appendix A). *p*-Values lower than or equal to 5 × 10^−8^ were considered genome-wide significant. To differentiate diabetes-specific and general genetic effects, the associations of the genetic variants with the apo-CIII proteoforms were investigated separately for people with or without type 2 diabetes, using linear regression models in SPSS.

### 4.5. Statistical Methods

The associations of apo-CIII glycosylation with lipid markers were analysed in the DiaGene Study using linear regression, separately for each apo-CIII proteoform. The proteoforms were considered the dependent variables. First, the analysis was performed with adjustments for age and sex; subsequently, a sensitivity analysis was performed for the use of lipid-lowering therapy (mostly statins and fibrates, combined as a binary variable). Next, this analysis was repeated for people with and without type 2 diabetes separately to investigate potential differences in medication effects. Since four separate proteoforms were analysed, the Bonferroni corrected *p*-Value for significance was applied as *p* < 0.0125 (0.05/4).

The genetic variants with a *p*-Value < 1 × 10^−6^ in the GWASs were selected for further statistical analyses with lipid markers and prevalence of type 2 diabetes. We performed multiple linear regression analyses of the association of the genetic variants with lipid markers HDL-c, non-HDL-c, LDL-c, total cholesterol, and triglycerides, adjusted for age and sex. The triglyceride concentrations were natural log-transformed for the statistical analyses; *p*-Values are shown for log-transformed triglyceride levels, while effect sizes are shown for non-transformed triglyceride concentrations. Next, we performed logistic regression analyses for the associations of the genetic variants with the prevalence of type 2 diabetes, adjusted for age and sex. The Bonferroni corrected threshold for significance was calculated as *p* < 0.0045 (0.05/11), as we analysed 11 single nucleotide polymorphisms (SNPs). Variants showing associations with a *p*-Value < 1 × 10^−6^ in the GWASs were investigated for associations with lipid markers in the Hoorn DCS Study. The results of DiaGene and Hoorn DCS of these variants with clinical outcomes were meta-analysed using beta, standard error, and number of participants of each cohort. Subsequently, SNPs were looked up in public GWAS databases Type 2 Diabetes Knowledge Portal [27] and IEU OpenGWAS [26] and in expression quantitative trait loci (eQTL) database eQTLGen [36] on 25 May 2022.

IBM SPSS 25.0 was used for all statistical analyses within the DiaGene study, R version 4.0.5 was used for statistical analysis in the Hoorn DCS and meta-analyses.

## Figures and Tables

**Figure 1 ijms-24-14844-f001:**
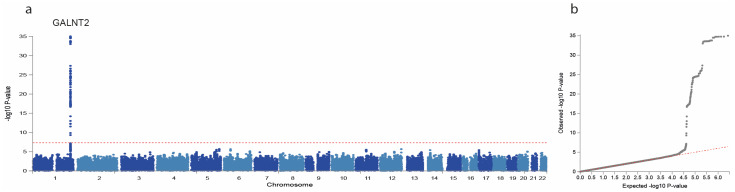
Manhattan and QQ plot of apo-CIII_0a_ genome-wide association study (GWAS). GWAS of apo-CIII_0a_ showing associations with alleles at the *GALNT2* locus on chromosome 1. Manhattan plot (**a**) showing significance of the association of each single nucleotide polymorphism (SNP) allele with apo-CIII_0a_ by plotting the -log10 of the *p*-Value against the genomic position. The horizontal red dotted line corresponds to the genome-wide significance threshold of *p* = 5 × 10^−8^. Quantile–quantile plot (**b**) is a plot of the observed -log_10_(p) against the expected -log_10_(p) under the null hypothesis of no association. Deviation above the red dotted y = x line indicates lower *p*-Values that would be expected to occur by chance and implies statistically significant association.

**Table 1 ijms-24-14844-t001:** General characteristics of the study populations.

	DiaGene	Hoorn DCS
Number of participants	2318	5409
Female sex, n (%)	1175 (50.7)	2414 (44.6)
Age, year, mean (±SD)	65.3 (9.5)	61.1 (11,0)
BMI, kg/m^2^, median (IQR)	28.0 (25.2–31.6)	29.4 (26.6–33.1)
Type 2 diabetes, n (%)	1572 (67.8)	5409 (100)
HDL-cholesterol, mmol/L, median (IQR)	1.2 (1.0–1.5)	1.2 (1–1.4)
Non-HDL cholesterol, mmol/L, mean (±SD)	3.4 (1.0)	3.8 (1.5)
LDL-cholesterol, mmol/L, mean (±SD)	2.8 (1.0)	2.9 (1.5)
Triglycerides, mmol/L, median (IQR)	1.4 (1.0–1.9)	1.7 (1.2–2.3)
Total cholesterol, mmol/L, mean (±SD)	4.7 (1.1)	5.0 (1.6)
Use of any lipid-lowering therapy, n (%)	1113 (50.3)	2276 (42.1)
Use of statins, n (%)	1090 (47.0)	2182 (40.3)
Use of fibrates, n (%)	33 (1.4)	68 (1.3)
Use of bile acid sequestrants, n (%)	1 (0.04)	6 (0.1)
Use of nicotinic acid and derivates, n (%)	5 (0.2)	7 (0.1)
Use of other lipid modifying agents, n (%)	56 (2.4)	56 (1.0)

Continuous data are presented as mean (and standard deviation) or median (and interquartile range) for normal and non-normal distributions, respectively. The distribution of the clinical variables was considered normal when Skewness and Kurtosis were within the range of −1 to +1. Percentages are calculated from the total number of participants. BMI, body mass index; HDL, high-density lipoprotein; LDL, low-density lipoprotein; n, number; SD, standard deviation; IQR, inter quartile range.

**Table 2 ijms-24-14844-t002:** Associations of apolipoprotein-CIII (apo-CIII) proteoforms with lipids and type 2 diabetes within the DiaGene study.

	HDL-c		
Proteoform	Beta	95% CI	*p*-Value
Apo-CIII_0a_	0.007	0.005 to 0.009	5.81 × 10^−9^
Apo-CIII_0c_	−0.002	−0.005 to 0.001	0.232
Apo-CIII_1_	−0.011	−0.015 to −0.007	3.24 × 10^−8^
Apo-CIII_2_	0.013	0.008 to 0.018	3.20 × 10^−7^
Apo-CIII_1_/apo-CIII_2_ ratio	−0.321	−0.422 to −0.221	4.00 × 10^−10^
	**Non-HDL-c**		
**Proteoform**	**Beta**	**95% CI**	** *p* ** **-Value**
Apo-CIII_0a_	0.004	0.003 to 0.004	1.48 × 10^−18^
Apo-CIII_0c_	0.002	0.001 to 0.003	3.09 × 10^−4^
Apo-CIII_1_	0.006	0.004 to 0.007	9.04 × 10^−16^
Apo-CIII_2_	−0.007	−0.009 to −0.006	3.60 × 10^−17^
Apo-CIII_1_/apo-CIII_2_ ratio	0.133	0.099 to 0.167	2.82 × 10^−14^
	**LDL-c**		
**Proteoform**	**Beta**	**95% CI**	** *p* ** **-Value**
Apo-CIII_0a_	0.004	0.003 to 0.005	3.86 × 10^−20^
Apo-CIII_0c_	0.001	0.000 to 0.002	0.041
Apo-CIII_1_	0.004	0.003 to 0.005	4.33 × 10^−8^
Apo-CIII_2_	−0.005	−0.007 to −0.003	5.74 × 10^−8^
Apo-CIII_1_/apo-CIII_2_ ratio	0.080	0.045 to 0.116	9.65 × 10^−6^
	**Total cholesterol**		
**Proteoform**	**Beta**	**95% CI**	** *p* ** **-Value**
Apo-CIII_0a_	0.004	0.003 to 0.005	4.37 × 10^−24^
Apo-CIII_0c_	0.002	0.001 to 0.003	0.003
Apo-CIII_1_	0.004	0.002 to 0.005	8.99 × 10^−9^
Apo-CIII_2_	−0.005	−0.007 to −0.004	4.17 × 10^−10^
Apo-CIII_1_/apo-CIII_2_ ratio	0.084	0.052 to 0.116	2.99 × 10^−7^
	**Triglycerides**		
**Proteoform**	**Beta**	**95% CI**	** *p* ** **-Value**
Apo-CIII_0a_	0.001	0.000 to 0.001	0.045
Apo-CIII_0c_	0.003	0.002 to 0.004	2.08 × 10^−8^
Apo-CIII_1_	0.008	0.007 to 0.009	9.66 × 10^−49^
Apo-CIII_2_	−0.011	−0.012 to −0.009	1.03 × 10^−49^
Apo-CIII_1_/apo-CIII_2_ ratio	0.225	0.193 to 0.257	2.26 × 10^−51^
	**Type 2 diabetes**		
**Proteoform**	**Beta**	**95% CI**	** *p* ** **-Value**
Apo-CIII_0a_	−29.59	−34.43 to −24.74	5.42 × 10^−33^
Apo-CIII_0c_	−0.963	−4.338 to 2.412	0.576
Apo-CIII_1_	1.275	−1.399 to 3.950	0.350
Apo-CIII_2_	−0.296	−2.377 to 1.786	0.781
Apo-CIII_1_/apo-CIII_2_ ratio	0.110	0.002 to 0.217	0.046

Adjusted for age and sex. Apo-CIII_0a_ normalized to all four proteoforms: apo-CIII_0a_, apo-CIII_0c_, apo-CIII_1_ and apo-CIII_2_. The sum of the glycoforms Apo-CIII_0c_, Apo-CIII_1_, and Apo-CIII_2_ was set to 1.0.

**Table 3 ijms-24-14844-t003:** Loci associated with at least one apo-CIII proteoform in the DiaGene study.

Proteoform	rsID	Chr	Pos	EA	RA	EAF	Beta	SE	*p*-Value	Locus
Apo-CIII_0a_	rs35498929	1	230286016	T	C	0.1249	−0.1223	0.02472	7.44 × 10^−7^	*GALNT2* (intronic)
	rs4846913	1	230294715	A	C	0.5964	−0.2027	0.01624	9.77 × 10^−36^	*GALNT2* (intronic)
	rs3213497	1	230416320	T	C	0.1282	−0.1271	0.02407	1.30 × 10^−7^	*GALNT2:RP5-956O18.3* (ncRNA exonic)
Apo-CIII_0c_	rs9378785	6	3316862	C	T	0.0508	−0.0918	0.01705	7.34 × 10^−8^	*SLC22A23* (intronic)
Apo-CIII_1_	rs10842926	12	27689893	C	G	0.0826	−0.0145	0.002928	7.07 × 10^−7^	*PPFIBP1* (intronic)
Apo-CIII_2_	rs67086575	2	27686480	G	A	0.1502	−0.0469	0.009473	7.32 × 10^−7^	*IFT172* (intronic)
	rs2493926	6	148614267	C	T	0.1112	0.0520	0.01058	9.12 × 10^−7^	*SASH1* (intronic)
	rs2481968	13	28567172	C	A	0.4874	0.0334	0.006647	4.90 × 10^−7^	*RN7SL272P* (intergenic)
	rs7175584	15	97500494	T	C	0.5638	−0.0339	0.00671	4.48 × 10^−7^	*RN7SKP181* (intergenic)
	rs10412211	19	13591542	T	G	0.4579	−0.0336	0.006713	5.62 × 10^−7^	*CACNA1A* (intronic)
Apo-CIII_1_/apo-CIII_2_ ratio	rs67086575	2	27686480	G	A	0.1502	0.0561	0.01133	7.37 × 10^−7^	*IFT172* (intronic)
	rs9462715	6	12368221	C	A	0.0571	−0.0826	0.01676	8.35 × 10^−7^	*RN7SKP293* (intergenic)
	rs2493926	6	148614267	C	T	0.1112	−0.0627	0.01266	7.46 × 10^−7^	*SASH1* (intronic)
	rs7175584	15	97500494	T	C	0.5638	0.0419	0.008028	1.80 × 10^−7^	*RN7SKP181* (intergenic)
	rs10412211	19	13591542	T	G	0.4579	0.0396	0.008031	8.24 × 10^−7^	*CACNA1A* (intronic)

An association was considered significant if the p value was lower than or equal to 5 × 10^−8^, the genome-wide significance threshold. Associations with *p*-values lower than or equal to 1 × 10^−6^ were considered at suggestive significance level. Chr, chromosome; Pos, position; EA, effect allele; RA, reference allele; EAF, effect allele frequency; SE, standard error; ncRNA, non-coding RNA.

**Table 4 ijms-24-14844-t004:** Significant associations of genetic variants with lipids.

HDL-c		*DiaGene*			*Hoorn DCS*			*Meta-Analysis*
rsID	EA	Beta	95% C.I.	*p*-Value	Beta	95% C.I.	*p*-Value	*p*-Value
rs4846913	A	0.006	−0.017 to 0.030	0.606	0.011	−0.001 to 0.023	0.076	0.065
rs9378785	C	−0.058	−0.113 to −0.003	0.040	0.005	−0.025 to 0.033	0.751	0.472
** Non-HDL-c **		** *DiaGene* **			** *Hoorn DCS* **			** *Meta-analysis* **
rsID	EA	Beta	95% C.I.	*p*-Value	Beta	95% C.I.	*p*-Value	*p*-Value
rs67086575	G	0.120	0.025 to 0.215	0.013	0.001	−0.056 to 0.059	0.961	0.207
rs2493926	C	−0.157	−0.266 to −0.048	0.005	0.030	−0.041 to 0.101	0.412	0.522
** LDL-c **		** *DiaGene* **			** *Hoorn DCS* **			** *Meta-analysis* **
rsID	EA	Beta	95% C.I.	*p*-Value	Beta	95% C.I.	*p*-Value	*p*-Value
rs67086575	G	0.109	0.018 to 0.201	0.019	−0.045	−0.097 to 0.006	0.086	0.707
rs2493926	C	−0.183	−0.288 to −0.078	0.001	0.035	−0.029 to 0.099	0.289	0.464
rs10842926	C	0.038	−0.084 to 0.160	0.547	0.089	0.016 to 0.162	0.016	0.016
** Total cholesterol **		** *DiaGene* **			** *Hoorn DCS* **			** *Meta-analysis* **
rsID	EA	Beta	95% C.I.	*p*-Value	Beta	95% C.I.	*p*-Value	*p*-Value
rs67086575	G	0.115	0.014 to 0.216	0.025	0.007	−0.051 to 0.065	0.817	0.190
rs2493926	C	−0.171	−0.286 to −0.055	0.004	0.017	−0.055 to 0.088	0.650	0.311
** Triglycerides **		** *DiaGene* **			** *Hoorn DCS* **			** *Meta-analysis* **
rsID	EA	Beta	95% C.I.	*p*-Value	Beta	95% C.I.	*p*-Value	*p*-Value
rs4846913	A	−0.063	−0.135 to 0.008	0.032	−0.052	−0.097 to −0.006	0.034	0.006
rs67086575	G	0.084	−0.014 to 0.183	0.017	0.084	0.023 to 0.145	2.20 × 10^−4^	1.07 × 10^−5^
rs10842926	C	−0.081	−0.212 to 0.051	0.061	−0.061	−0.147 to 0.025	0.185	0.039
rs2481968	C	−0.076	−0.146 to −0.007	0.025	0.017	−0.028 to 0.062	0.759	0.391

Adjusted for age and sex.

## Data Availability

The data presented in this study are available on request from the corresponding author. The data are not publicly available due to privacy regulations.

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
