# Peer review of "Apolipoprotein-CIII O-Glycosylation, a Link between GALNT2 and Plasma Lipids"

_ijms, 2023, doi:10.3390/ijms241914844_

Round 1

Reviewer 1 Report

This is very well written original research report. The authors demonstrate that variants of the GALNT2-gene affect apo-CIII O-glycosylation and lipid  metabolism, and have an impact on the risk of type 2 diabetes.

Minor comment:

The GALNT2-gene encodes the GalNAc-T2 enzyme that covalently links N-acetylgalactosamine (GalNAc) to proteins. The sentence in line 345 (The genetic variation...) does not make sense.

Author Response

Thank you very much for taking the time to review this manuscript. We changed the sentence in line 345 as you suggested. Please find the corresponding revisions in track changes in the re-submitted files.

Reviewer 2 Report

Annemieke Naber et al. Investigated the abundance of the different apoCIII O-glycosylated isoforms  in two cohorts consisting of healthy controls and T2DM patients. The major conclusion of this study is that i) apoCIII glycosylation is affected by polymorphisms (mutations) of the  GALNT2 gene and ii) plasma lipid and lipoprotein  levels are affected by the ratio of different apoCIII isoforms.

COMMENTS

This is a pure observational study with little link to mechanistic pathways and the results are partly controversial to previously published work. Nevertheless I  consider the data of great interest and an important basis of forthcoming work.

As the authors partly point out in their Discussion, the results of this work open a large amount of questions that might be answered by analyzing the data in view of numerous questions:

1.    One major obvious point is the ”hen : egg” question, whether apoCIII glycosylation is causally related to T2DM or vice versa. 

2.    Another question to be answered is whether apoCIII glycosylation affects directly the activity of LPL or HL? If so, the different TG values in the plasma of probands with different rations of apoCIII glycosylation or GALNT2 mutations might be explained.

3.    Also, as the authors point out there are several additional proteins and enzymes involved in lipoprotein metabolism ( ANGPTL3, LCAT, PLTP…) are also glycosylated by  GALNT2. It might be interesting to know whether the glycosylation of these latter proteins parallels that of apoCIII in T2DM patients.

Specific Questions:

1.    The authors point out that absolute plasma apoCIII concentrations were not available from the two cohorts. How then did they calculate the ratio of the different apoCIII isoforms? Just by comparing the signals in their LC-MS analysis? If so, what did they use as reference material, how were samples hydrolyzed and what was the precision and CV of the method. A more detailed description will be welcome.

2.    Influence of drugs: The authors mention only statin treatment of part of their probands. How about anti-diabetic drugs? Is there any information available and if so how did that affect the apoCIII ratios?

In summary, the manuscript is interesting and might be the basis of further more mechanistic investigations. The reply to some of the questions mentioned above will be welcome.

Reviewer 3 Report

The authors aimed to determine how apo-CIII glycosylation affects lipid traits and type 2 diabetes prevalence, and to investigate the genetic basis of these relations with a GWAS on ApoCIII glycosylation. They conducted GWAS on the four apolipoprotein-CIII proteoforms in the DiaGene study in people with and without type 2 diabetes (n=2,318). They investigated the relations of the identified genetic loci and apolipoprotein-CIII glycosylation with lipids and type 2 diabetes. The associations of the genetic variants with lipids werereplicated in the Diabetes Care System (n=5,409). Rs4846913-A, in the GALNT2-gene, was associated with decreased apo-CIII0a. This variant was associated with increased high-density lipoprotein cholesterol and decreased triglycerides, while high apo-CIII0a was associated with raised HDL-cholesterol and triglycerides. Rs67086575-G, located in the IFT172-gene, was associated with decreased apo-CIII2 and with hypertriglyceridemia. In line, apo-CIII2 was associated with low triglycerides. On a genome-wide scale, they confirmed that the GALNT2-gene plays a major role in O-glycosylation of apolipoprotein-CIII, with subsequent associations with lipid parameters. They newly identified the IFT172/NRBP1 region, in literature previously associated with hypertriglyceridemia, as involved in apolipoprotein-CIII sialylation and hypertriglyceridemia. They concluded that these results link genomics, glycosylation and lipid metabolism, and represent a key step towards unravelling the importance of O-glycosylation in health and disease.

The study is well organized and well presented reporting some important novel data on apolipoprotein-CIII sialylation.

Comments:

·         Lipid lowering medication mentioned in Table 1 should be specified, since it may alter the results.

·         Tables: Lists of abbreviations should be added.

·         Table 2. line Proteoform: beta instead of Beta

·         Quality of Figure 1 could be improved.

Reviewer 4 Report

This is an interesting study of the impact of protein glycosylation on function with a genetic explanation of why certain effects happen. It is a constant query in lipid clinics what to do about patients with high HDL because we do not know whether it is beneficial or not., Therefore, information that may explain HDL could be very helpful in determining treatment…

This is a very well conducted study and it is presented very well. The GWAS graph shows convincingly that there are several SNPs associated with the GALNT gene that link with glycosylation. Overall, I feel that this manuscript is one of those very rare events where the science is well presented and the manuscript does not need any changes!

Author Response

Thank you very much for your compliment and for taking the time to review this manuscript.